# Effect of Curing Conditions on Microstructure and Pore-Structure of Brown Coal Fly Ash Geopolymers

**Chamila Gunasekara [1],\* , Rahmat Dirgantara [2], David W. Law [1] and Sujeeva Setunge [1]**

[1]  Civil and Infrastructure Engineering, School of Engineering, RMIT University, Melbourne, VIC 3000, Australia
[2]  Civil Engineering Department, Faculty of Computer and Engineering, Universitas Harapan, Medan 20216, Indonesia
\*  Correspondence: chamila.gunasekera@rmit.edu.au; Tel.: +61-3-9925-0321

**Abstract:** This study reports the effect of heat curing at 120 °C on the geopolymeric reaction and strength evolution in brown coal fly ash based geopolymer mortar and concrete. Moreover, an examination of this temperature profile of large size geopolymer concrete specimens is also reported. The specimen temperature and size were observed to influence the conversion from the glassy (amorphous) phases to the crystalline phases and the microstructure development of the geopolymer. The temperature profile could be divided into three principal stages which correlated well with the proposed reaction mechanism for class F fly ash geopolymers. The geopolymerisation progressed more rapidly for the mortar specimens than the concrete specimens with 12 to 14 h providing an optimum curing time for the 50 mm mortar cubes and 24 h being the optimum time for the 100 mm concrete cubes. The 50 mm and 100 mm concrete specimens' compressive strengths in excess of 30 MPa could be obtained at 7 days. The structural integrity was not achieved at the center of 200 mm and 300 mm concrete specimens following 24 h curing at 120 °C. Hence, the optimal curing time required to achieve the best compressive strength for brown coal geopolymer was identified as being dependent on the specimen size.

**Keywords:** geopolymer; brown coal fly ash; compressive strength; heat curing; porosity; X-ray computed tomography

## 1. Introduction

Conventional concrete produced by using Portland Cement (PC) as the main binder releases between 0.7 to 1.0 kg of $CO_2$ per 1 kg of cement production [1,2], and as a result, is responsible for between 5 to 7% of anthropogenic $CO_2$ emissions [1,2] which contributes to the global warming. Thus, cement usage for concrete production is not sustainable as the production process is energy intensive and expensive. The Asia-Pacific population has grown to reach nearly 4.5 billion, while urbanization in most of the region is increasing. In response to this growth, the number of power plants across the region has increased significantly, bringing power to growing populations and industrial centers. In addition, the Asia-Pacific region's electricity is largely derived from coal, which represents over half of the power mix [3]. Hence, the current emphasis is on developing novel low-$CO_2$ binders, such as geopolymers, to meet the demand for concrete.

Geopolymer concrete can be manufactured using a wide range of aluminate-silicate source materials, including different types of fly ash, blast furnace slag and metakaolin [4–9]. There are two types of fly ash readily available, high-calcium (class C) and low-calcium (class F) fly ash [10]. The class F fly ash is produced from burning anthracite and bituminous coals, whereas class C fly ash is

produced from subbituminous and lignite coal. Due to the high aluminosilicates and low calcium, Class F fly ash is the preferred material for the production of geopolymer concrete.

In Australia, lignite coal is referred to as brown coal, whereas in Europe, both subbituminous coal and lignite coal are called brown coal [11]. The state of Victoria in Australia utilizes approximately 70 million tonnes of brown coal deposits mined from the Latrobe Valley in order to satisfy nearly 90% of the annual electricity consumption. The annual worldwide production of brown coal was estimated to be approximately 950 million tonnes in 2008, and Australia's brown coal production is responsible for 9% of the total production [11]. Further, Australia's recoverable brown coal of 44.2 billion tonnes is approximately 19% of the world's recoverable resources, with 93% deposited in the Latrobe Valley, Victoria [12]. At present in Australia, brown coal fly ash is not used as a binder in concrete due to the composition of the coal. As brown coal fly ash has a high sulphur content (above 5%), this means it cannot be classified as either Class F or Class C according to ASTM C618 [10], resulting in large quantities of brown coal fly ash deposited in storage lagoons with subsequent environmental costs [13,14].

The suitability of brown coal fly ash for geopolymer production has been investigated by Dirgantara et al. [15] who reported that the maximum 28-day compressive strength of the geopolymer mortar obtained was 57 MPa, which is comparable to a low-calcium (Class F) fly ash–based geopolymer. The leaching of heavy metals in blended brown coal fly ash-metakaolin geopolymer binders was examined by Bankowski et al. [16], however, they did not expand this investigation to assess the engineering properties. The results showed that the geopolymer was effective at reducing the leaching rates of calcium, arsenic, selenium, strontium and barium. Tennakoon et al. [6] developed a blended brown coal fly ash geopolymer with low calcium fly ash and blast furnace slag. The study showed that brown coal fly ash with higher aluminium content improved the rate of reaction of the geopolymer.

Bakharev [17] reported that the geopolymers are ceramic materials that are produced by alkali activation of aluminosilicate raw materials, which are transformed into reaction product by polymerisation in a high pH environment and hydrothermal conditions at relatively low temperatures. It was further noted that ambient temperature (23 ± 1 °C) curing does not achieve a final setting for 100% low calcium fly ash based geopolymer concrete, with a number of researchers studying the activation of low calcium (class F) fly ash by alkalis utilizing heat curing in the range 60–80 °C [18–21]. Sivasakthi et al. [22] reported that micro silica–fly ash blended geopolymer concrete prepared using a hybrid solution of sodium silicate and sodium hydroxide cured at 80 °C for 24 h and 7 days at ambient temperature (23 ± 1 °C) is thermally stable up to 800 °C and that the compressive strength obtained satisfied the requirement of building material with high-temperature resistance in modern construction and the ceramic industries.

You et al. [23] examined the technical feasibility of geopolymer synthesis from the coal fly ash with high iron oxide and calcium oxide contents, and concluded that it is possible to improve the economics of geopolymer production by varying the material use while not impairing the performance of geopolymer. The curing temperature and curing time for geopolymer concrete have been reported to vary significantly based on the source of the fly ash. Gunasekara et al. [24] used heat curing of 80 °C temperature for 24 h to produce geopolymer concrete using low calcium, Class F fly ash obtained from four different power stations in Australia. Chindaprasirt al. 2013 [25] produced geopolymer concrete using high calcium, Class C fluidized bed combustion fly ash, cured at 60 °C for 24 h. Phoo-Ngernkham et al. [26] showed that the 28-day compressive strength of 15–35 MPa can be obtained using alkali-activated high calcium fly ash concrete cured at ambient temperature (23 ± 1 °C). Further, Diaz et al. [27] used three high calcium, Class C fly ashes obtained from power plants around the United States in order to produce geopolymer concrete while it cured for 3 days at 60 °C. Palomo et al. [28] produced geopolymer concrete using law calcium Class F fly ashes cured at 65 °C and 85 °C for 24 h, and showed the importance of heat curing, where a significant increase in strength was observed at 85 °C as compared to 65 °C. Moreover, Bakharev [17] developed geopolymer concrete using two Australian low calcium Class F fly ashes cured at 100 °C for 24 h. In the case of brown coal fly ash,

Dirgantara et al. [15] observed the alkali activation of brown coal fly ash obtained from La Trobe Valley, Victoria, Australia required heat curing of 120 °C for 24 h to optimize the strength development and that the application of lower curing temperatures gave significantly lower compressive strengths.

## 2. Research Significance

Heat curing has been identified as accelerating the alkali-fly ash chemical reaction that occurs in 100% fly ash based geopolymer [18–20], with the temperature observed to significantly affect the structural transition from the amorphous to the crystalline and the microstructure development of the geopolymer. Further, research has shown that a curing temperature of 120 °C is required for brown coal fly ash [15]. However, the effect of this curing temperature on the microstructure development is unclear, in particular, with regard to water evaporation and the size effects within the specimen. This paper reports a study on the effect of elevated heat curing on the temperature profile and microstructure development in brown coal fly ash geopolymer mortar and concrete specimens.

## 3. Experimental Procedure

### 3.1. Materials Used

The brown coal fly ash used in this study was obtained from the Loy Yang power plant in the La Trobe Valley, Victoria, Australia. The brown coal fly ash supplied was obtained directly from the electrostatic precipitators with no treatment or time in storage ponds prior to being used in concrete production. Table 1 shows the chemical composition of the ash, determined by Bruker Axs S4 Pioneer X-ray fluorescence (XRF) instrument (Bruker Corporation, Massachusetts, United States). The significant variation in the CaO content has been observed in brown coal both in Australia and from other countries [13,29]. This has included CaO concentrations in excess of 2.25% and as such is not regarded as untypical. The particle size distribution was obtained using Malvern Mastersizer analyzer (Malvern Panalytical Ltd, Malvern, United Kingdom). The amorphous content and crystallinity of Loy Yang fly ash was determined using Bruker Axs D8 ADVANCE Wide Angle X-Ray Diffraction (XRD) instrument (Bruker Corporation, Massachusetts, United States) and presented in Table 2. The XRD measurement was performed at 40 kV, Cu K$\alpha$ = 1.54178 Å wavelength; scanning range 2 theta in 5°–95°. The sample holders were filled using the front-loading procedure. The data obtained from XRD were interpreted using Bruker–DIFFRAC.EVA software and a Rietveld analysis [30,31] in order to quantify the amorphous and crystalline percentage. The fly ash surface area was determined using Brunauer Emmett Teller method by $N_2$ absorption [32,33].

**Table 1.** The chemical composition.

| Brown Coal Fly Ash | By Weight (%) | | | | | | | | | | |
|---|---|---|---|---|---|---|---|---|---|---|---|
| | $SiO_2$ | $Al_2O_3$ | $Fe_2O_3$ | CaO | $K_2O$ | $TiO_2$ | $P_2O_5$ | MgO | $Na_2O$ | $SO_3$ | MnO |
| Loy Yang | 47.52 | 17.29 | 5.98 | 2.25 | 0.50 | 1.26 | 0.74 | 4.63 | 6.26 | 13.03 | 0.54 |

**Table 2.** The physical and mineralogical properties.

| Properties Investigated | | Loy Yang |
|---|---|---|
| BET Surface Area, ($m^2$/kg) | | 279 |
| % passing at | 10 microns | 6.5 |
| | 20 microns | 19.7 |
| | 45 microns | 45.1 |
| | 75 microns | 62.0 |
| Amorphous (%) | | 62.5 |
| Crystalline (%) | | 37.5 |
| Unburnt carbon content (%) | | 1.43 |

The alkali activator used contained a mixture of commercially available sodium hydroxide (15 M) solution from Science Supply Australia and the sodium silicate solution had a specific gravity of 1.53 and alkaline modulus = 2, where $Na_2O$ = 14.7%, $SiO_2$ = 29.4% and $H_2O$ = 55.9% by mass [5,34]. Both the coarse and fine aggregate were prepared in accordance with AS 1141.5 [35]. The river sand in uncrushed form (specific gravity of 2.5 and a fineness modulus of 3.0) was used as a fine aggregate. The granite crushed aggregate of two-grain sizes, 7 mm of 2.58 specific gravity and 1.60% water absorption and 10 mm of 2.62% specific gravity and 0.74% water absorption, were used as coarse aggregates. It is noted that the coarse aggregate was selected to produce laboratory-scale concrete specimens and is comparable with the other studies [34,36,37].

### 3.2. Mix Designs

The mix design used for the brown coal fly ash geopolymer mortar and concrete are based on the optimum mix composition from previous research [15], Table 3. It is noted that the mix design for the concrete is based on the mix proportions of the geopolymer mortar. The sand proportion was substituted by a combination of aggregates, i.e., 43% of sand, 38% of 10 mm aggregate and 19% of 7 mm aggregate. The same water to solid (w/s) ratio of optimum mortar mix (i.e., w/s = 0.37) was used for the concrete mix design. In the w/s ratio, the quantity of water contained in the mix is defined as the sum of water contained in the sodium silicate and the sodium hydroxide solution and added water, whereas the quantity of the solid is the sum of the mass of fly ash and the solid contained in the alkaline activator solution.

**Table 3.** The optimum mix design ($kg/m^3$).

| Loy Yang Geopolymer | Fly Ash (kg) | Aggregates (kg) | | | Activator (kg) | |
|---|---|---|---|---|---|---|
| | | Sand | 7 mm | 10 mm | $Na_2SiO_3$ (Liquid) | NaOH (15 M) |
| Mortar | 279 | 1677 | | | 363 | 21 |
| Concrete | 279 | 721 | 319 | 637 | 363 | 21 |

### 3.3. Specimen Preparation and Curing

In the brown coal fly ash geopolymer mortar preparation, fly ash and sand were mixed using a 5-L Hobart mixer for 4 min. The alkaline activator solution, consisting of sodium hydroxide and sodium silicate solutions as well as additional water, was prepared manually 15 min prior to mixing. The mortar mixture was produced by the addition of pre-prepared alkaline solution to the dry mix of fly ash and sand, and mixing for 4 min at a speed of 140 rpm followed by an additional mixing for 2 min at 285 rpm in a Hobart mixer. After mixing, the geopolymer mortar was poured into the 50 mm cubic Teflon moulds and vibrated using a vibration table at a speed of 225 rpm for 30 s. In the geopolymer concrete preparation, the fly ash, coarse aggregate and sand were mixed using a 60-L concrete mixer (speed of 290 rpm) for 4 min, the pre-mixed alkali activator solution together with the extra water were added and mixed for further 8 min. The geopolymer concrete mix was placed into 50 mm, 100 mm, 200 mm and 300 mm cubic timber moulds, and vibrated using a vibration table at a speed of 225 rpm for 1–2 min to remove air bubbles.

The temperature probes were placed at fixed depths in the specimens in order to monitor the temperature profile of the mortar and concrete when exposed to heat curing as experienced under normal oven conditions, Figure 1. The temperature probes were monitored through a data logger attached to a computer. In mortar specimens, a single thermocouple was positioned in the center. In the 100 mm cube, the concrete sample probes were positioned along the centerline at the 25, 50, and 75 mm distance from the side and from the top surface. In the 200 and 300 mm cubes, the probes were positioned at 25 mm intervals from the side along the centerline and the depth increased by 25 mm from the top surface respectively.

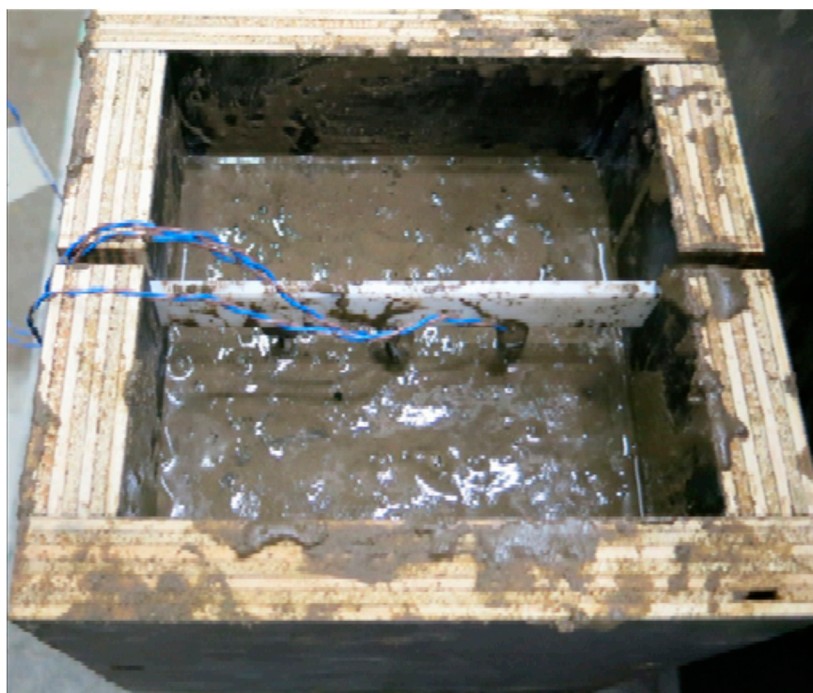

**Figure 1.** The concrete specimen (100 mm cube) with thermocouples embedded.

Both mortar and concrete specimens were covered to prevent moisture loss and left for 24 h under laboratory conditions (24 °C and 70% relative humidity) before heat curing. A curing temperature of 120 °C was employed with a range of curing times between 4 and 14 h at 2-h time intervals for the mortar and concrete specimens with a further concrete specimen monitored for 24 h. This curing temperature was obtained in the mix optimization process of 100% brown coal fly ash geopolymer mortar, based on 28-day compressive strength [15]. The geopolymer specimens were evenly placed at the center of the dry oven to ensure uniform heating of all specimens. The specimens (either mortar or concrete) were removed from the oven after the specified curing time. Following heat curing, all specimens were taken out from the oven, then demoulded and stored in laboratory conditions.

### 3.4. Testing

The compressive strength of mortar specimens was tested using a Technotest concrete testing machine in accordance with AS 1012.9 [38] with a loading rate of 0.34 N/mm$^2$/s. The compressive strength test on concrete specimens was performed with a MTS machine with a loading rate of 20 MPa/min according to AS 1012.9 [38]. In each mix design, three specimens were tested for the 7-day compressive strength and the mean value was reported.

The scanning electron microscope (SEM) with backscatter electron imaging (15 kV of energy) (Olympus Corporation, Tokyo, Japan) and an energy-dispersive X-ray spectroscope (EDS) were used to examine the microstructure and the elemental distribution of brown coal fly ash geopolymer mortar, respectively. All tested specimens were 5 mm in height and 12 mm in diameter, and carbon coated. The pore structure was investigated using mercury intrusion porosimetry (MIP) and X-ray computed tomography (CT) [39]. The porosity distribution analysis was conducted using an Auto pore IV 9500 (V1.09) micrometrics mercury intrusion porosimeter. Prior to the analysis, the geopolymer mortar monoliths (1 cm × 1 cm × 2 cm) were dried in an oven at 105 °C (221 °F) for 48 h.

The *X*-ray CT was used to investigate pore connectivity and pore-structure of the brown coal fly ash geopolymer concrete. The duplicate 10 mm × 10 mm × 50 mm specimens were scanned at 20 μm resolution at 180 kV and 100 μA, and 1000 images recorded during a complete rotation. The samples (prepared with same concrete mix) were scanned at 4, 8, 12 and 24 h to identify changes in the

pore structure with time. Finally, VGStudioMax 2.2 software (Volume Graphics GmbH, Heidelberg, Germany) was used to analyse the porosity of tested concrete specimens.

## 4. Results

### 4.1. Heat Curing vs. Specimen Size

Figure 2 shows the temperature variation in the center of the geopolymer concrete cubes, 50 mm, 100 mm, 200 mm and 300 mm together with the temperature profile of the 50 mm geopolymer mortar cube. It is interesting to note that 50 mm geopolymer mortar and concrete cubes showed an almost identical temperature profile. Three distinct phases within the heat curing, in both 50 mm specimens, were identified which corresponded to 2–6 h, 6–10 h and 10–14 h. The temperature profile shows that by 2 h, the internal temperature had reached approximately 105 °C, between 2 to 6 h there was then a steady rise in temperature to 108 °C, and from 6 to 10 h a more rapid rise was observed to 120 °C. This is followed by a slow rise in temperature to 14 h where the temperature inside (center) of the cube was just below the oven temperature of 120 ± 5 °C.

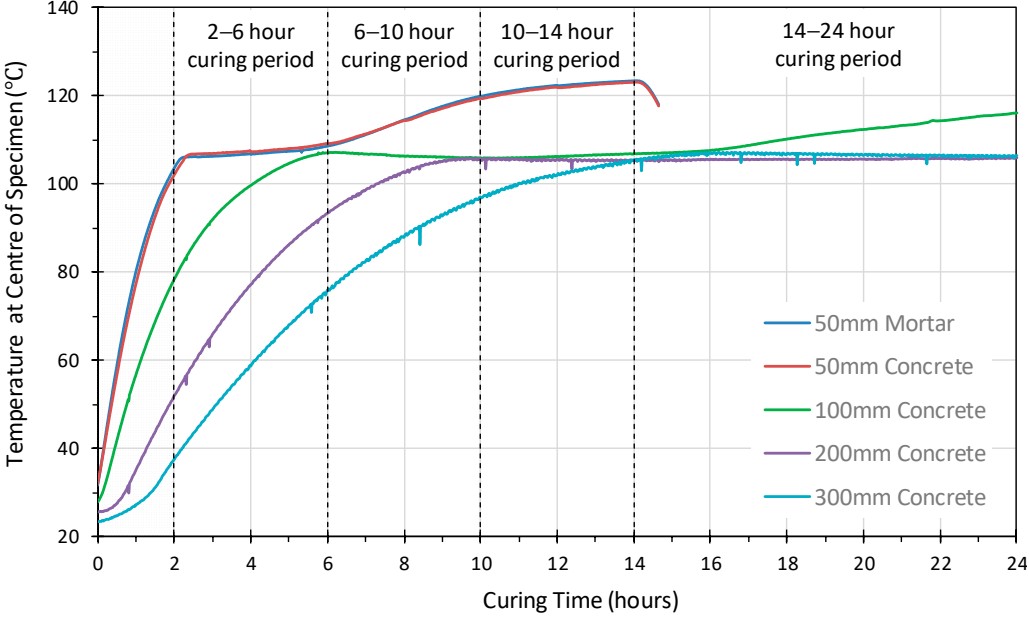

**Figure 2.** The temperature development at the center of the geopolymer mortar and concrete specimens.

However, the temperature profile of the 50 mm concrete cube is very different to the profile at the center of the larger geopolymer concrete cubes. The temperature profiles of the larger cubes exceeded 100 °C after different curing periods. This was after 4 h for the 100 mm cube, 7 h for the 200 mm cube and 11 h for 300 mm cube. All specimens subsequently rose to above 105 °C during the first 14 h. Furthermore, the 100 mm, 200 mm and 300 mm cubes all showed a slight drop in temperature of 1–2 °C before remaining constant in the 105 to 107 °C range. The 100 mm cube displayed a steady rise in temperature from 16 h until 24 h, the temperature exceeding 115 °C at this point. Both the 200 mm and 300 mm cubes maintained a relatively constant temperature in the range 105–107 °C until 24 h.

Figures 3 and 4 illustrate the temperature variation versus the depth within the 100 mm, 200 mm and 300 mm concrete specimens. Similar profiles were observed for all the probes in the 100 mm cube. The 75 mm sensor achieved slightly higher temperatures earlier than those at 25 mm and 50 mm. The internal temperature exceeded 100 °C at approximately 4 h, by 6 h, it had reached 107 °C, and then dropped by 1–2 °C and remained stable for approximately 4 h before starting to increase again after 10 h curing and continued to rise until the 24 h heat curing period was complete. This would suggest that a uniform temperature was achieved throughout the specimen. On the other hand, the

temperature profiles of the 200 mm and 300 mm specimens, Figures 4 and 5, show that the internal temperature exceeded 100 °C between 6 to 8 h for 200 mm cube and between 7 to 13 h for 300 mm cube. In both cubes, the probes close to the surface reached 100 °C before those at the centre of the specimen, indicating that a uniform temperature was not achieved throughout the specimen.

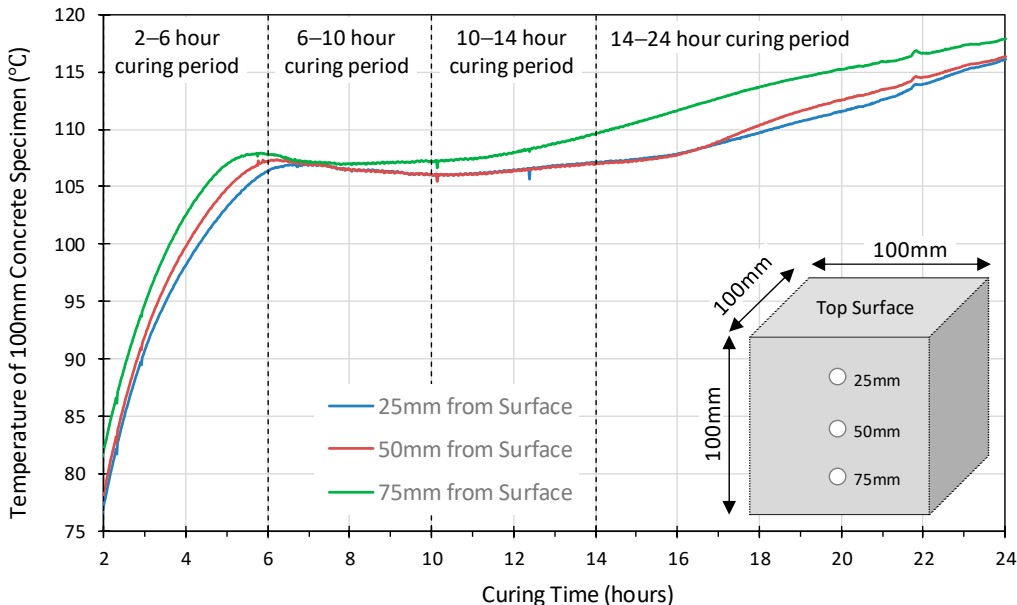

**Figure 3.** The temperature development versus the depth of 100 mm geopolymer concrete cube.

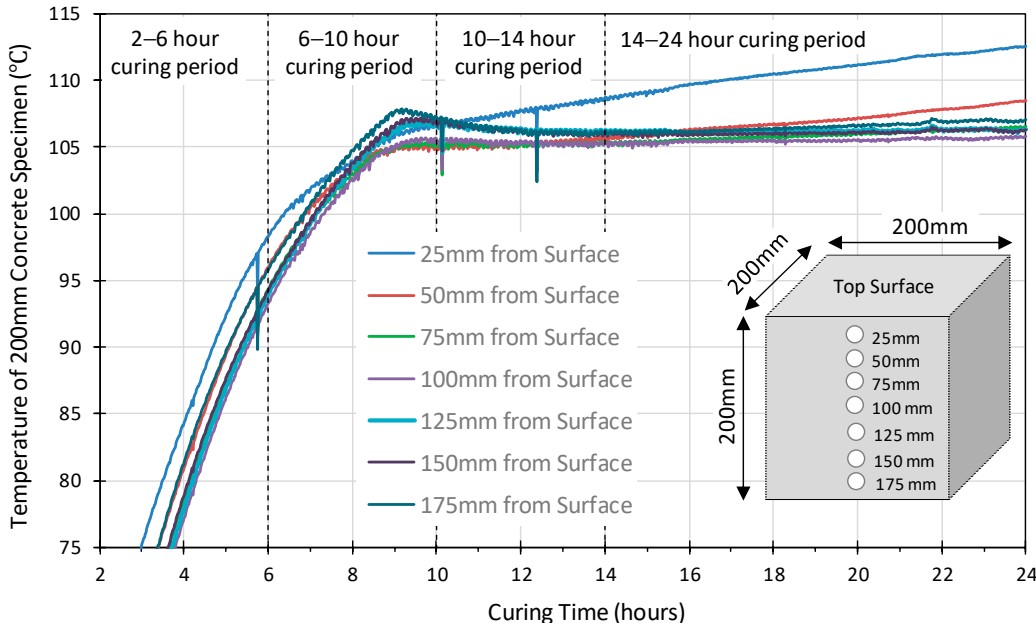

**Figure 4.** The temperature development versus the depth of 200 mm geopolymer concrete cube.

The 200 mm cube profile shows a drop followed by a slight steady rise in the temperature between 105 °C and 110 °C up to 24 h, for all locations other than at 25 mm which show a constant steady rise throughout. This is identified with the structural re-organization and gelation phase at the conclusion of the dissolution phase. The drop in temperature is hypothesised as relating to the energy required during this phase. All the sensors in the 300 mm cube also show a slight drop in the temperature, as observed in both the 100 mm and 200 mm cubes, but no increase is observed in any other than at 50 mm by the end of the 24-h curing period. The profiles indicate that in the larger 200 mm and 300 mm specimens, the heat transfer is significantly reduced with significantly lower temperatures within the

center of the specimens than the surface 50 mm of the concrete. Indeed, the structural integrity was not achieved by the 200 mm and 300 mm specimens, which would suggest that the geopolymerisation reaction has not been completed within the 24-h curing period for these specimens.

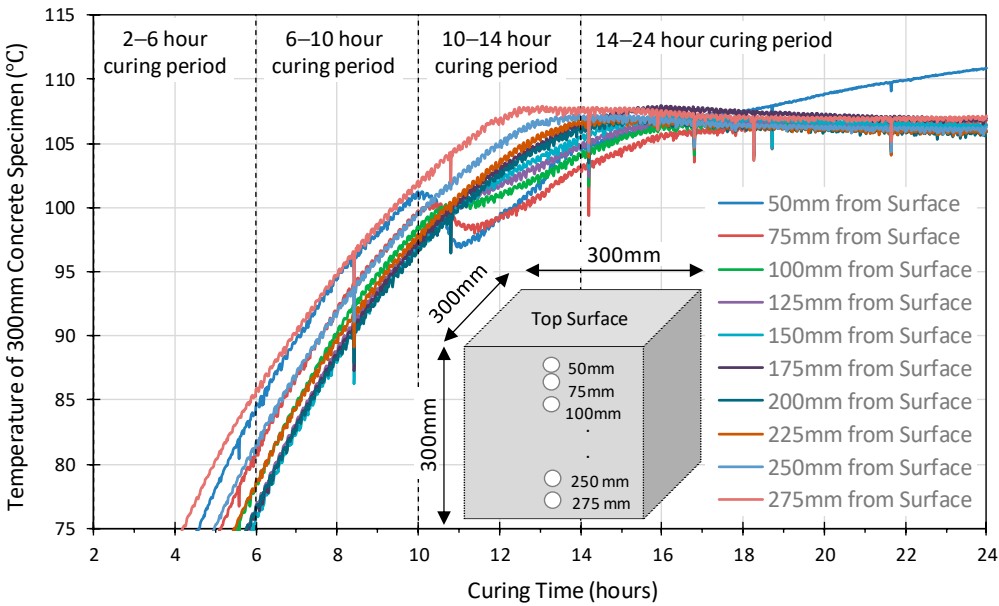

**Figure 5.** The temperature development versus the depth of 300 mm geopolymer concrete cube.

### 4.2. Compressive Strength vs. Curing Time

The compressive strength in brown coal fly ash geopolymer mortar (50 mm cubes) and brown coal fly ash geopolymer concrete (both 50 mm and 100 mm cubes) at 8, 10, 12 and 14 h curing are shown in Table 4 and Figure 6. It is noted that the standard size laboratory scale concrete and mortar specimens were tested for compressive strength. The data shows an increase in strength with curing time, consistent with that observed in previous studies [28].

**Table 4.** The effect of curing time on 7-day compressive strength.

| Curing Time | Mean Compressive Strength (MPa) | | |
|---|---|---|---|
| | Mortar (50 mm Cube) | Concrete (50 mm Cube) | Concrete (100 mm Cube) |
| 4 h | 4.60 ± 1.05 | 3.05 ± 0.65 | 1.10 ± 0.45 |
| 6 h | 23.00 ± 6.75 | 14.55 ± 2.75 | 8.75 ± 1.85 |
| 8 h | 33.55 ± 6.75 | 22.85 ± 1.15 | 10.95 ± 1.40 |
| 10 h | 33.65 ± 8.15 | 26.40 ± 2.05 | 12.55 ± 2.55 |
| 12 h | 42.10 ± 5.85 | 34.95 ± 2.65 | 14.25 ± 3.95 |
| 14 h | 31.00 ± 3.55 | 33.70 ± 3.05 | 18.10 ± 3.95 |
| 24 h | | | 30.85 ± 3.45 |

The 7-day compressive strength of geopolymer mortar (50 mm cube) and geopolymer concrete (100 mm cube) varied from 4.6–31.0 MPa (4–14 h) and 1.1–30.85 MPa (4–24 h), respectively, with the mortar showing higher strength than the concrete at all curing times. The maximum 7-day strength obtained for the mortar and concrete, are 42.1 MPa and 30.85 MPa, respectively. These are comparable with the strengths achieved by a number of Class F fly ashes [33,40,41]. It is interesting to note that geopolymer mortar obtained its highest compressive strength after 12 h but subsequently reduced in strength from 12 to 14 h, losing 11 MPa or 26% of the strength. On the other hand, the 100 mm concrete specimen strength increased by 4 MPa in the same period and gained 16 MPa, more than doubling in strength, from 12 to 24 h.

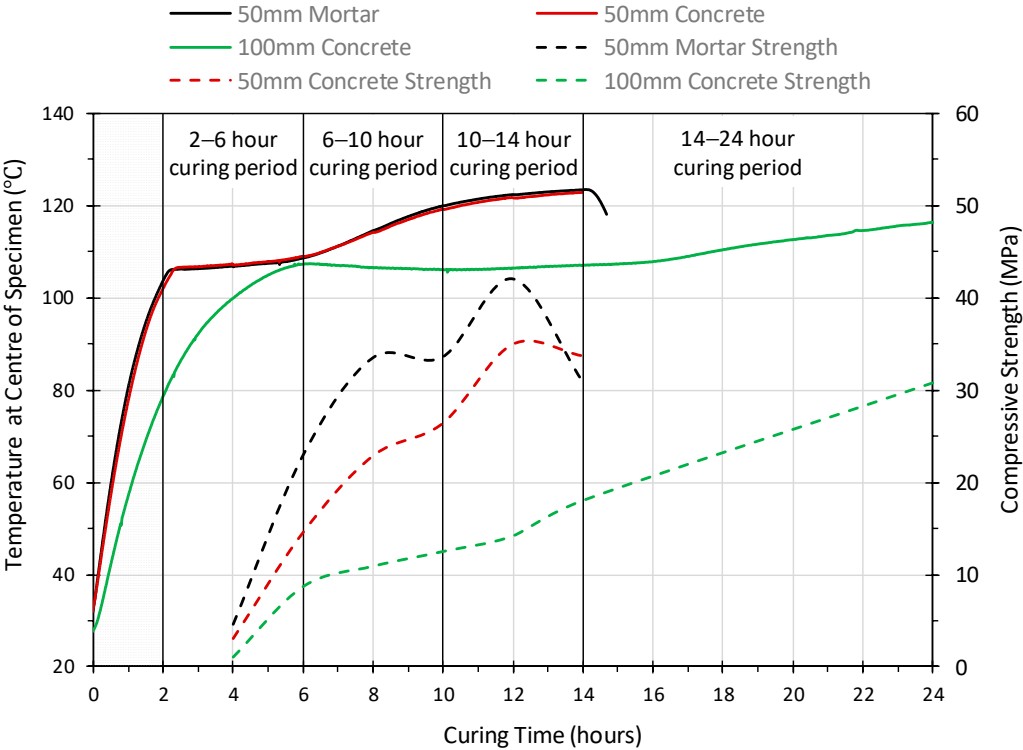

**Figure 6.** The compressive strength development profile of geopolymer mortar and concrete.

### 4.3. Microstructure

Figure 7 shows the microstructure of the 50 mm geopolymer mortar at 8, 10, 12 & 14 h. At 8 h, the geopolymer displays a porous, less compacted structure with a number of unreacted fly ash grains present. The images at 10 and 12 h show a more compact homogeneous structure. Some unreacted fly ash grains were still observed at 10 h, but by 12 and 14 h few remained, indicating that the gelation process reached a conclusion. Micro cracking is also observed at 10, 12 and 14 h, with a significant increase in the number and size of the micro cracks at 14 h. This is attributed to the extended curing at high temperatures which has been observed to increase cracking [42].

The changing microstructure of the 50 mm geopolymer concrete cube at 8, 10, 12 and 14 h is illustrated in Figure 8. Similar to the mortar specimens, the geopolymer concrete shows a porous less compacted structure with unreacted/partially reacted fly ash spheres at 8 h. However, with the increase in the curing time, less unreacted fly ash and more geopolymeric gel was observed. This resulted in a more compacted homogeneous aluminosilicate gel structure evident at 10, 12 and 14 h. It was also noted that a significant reduction in the number of micro cracks was observed after 14 h of curing compared to that observed after 12 h of curing. This is in contrast to the microstructure of the mortar, which showed an increase in the number of micro-cracks between 12 and 14 h of curing.

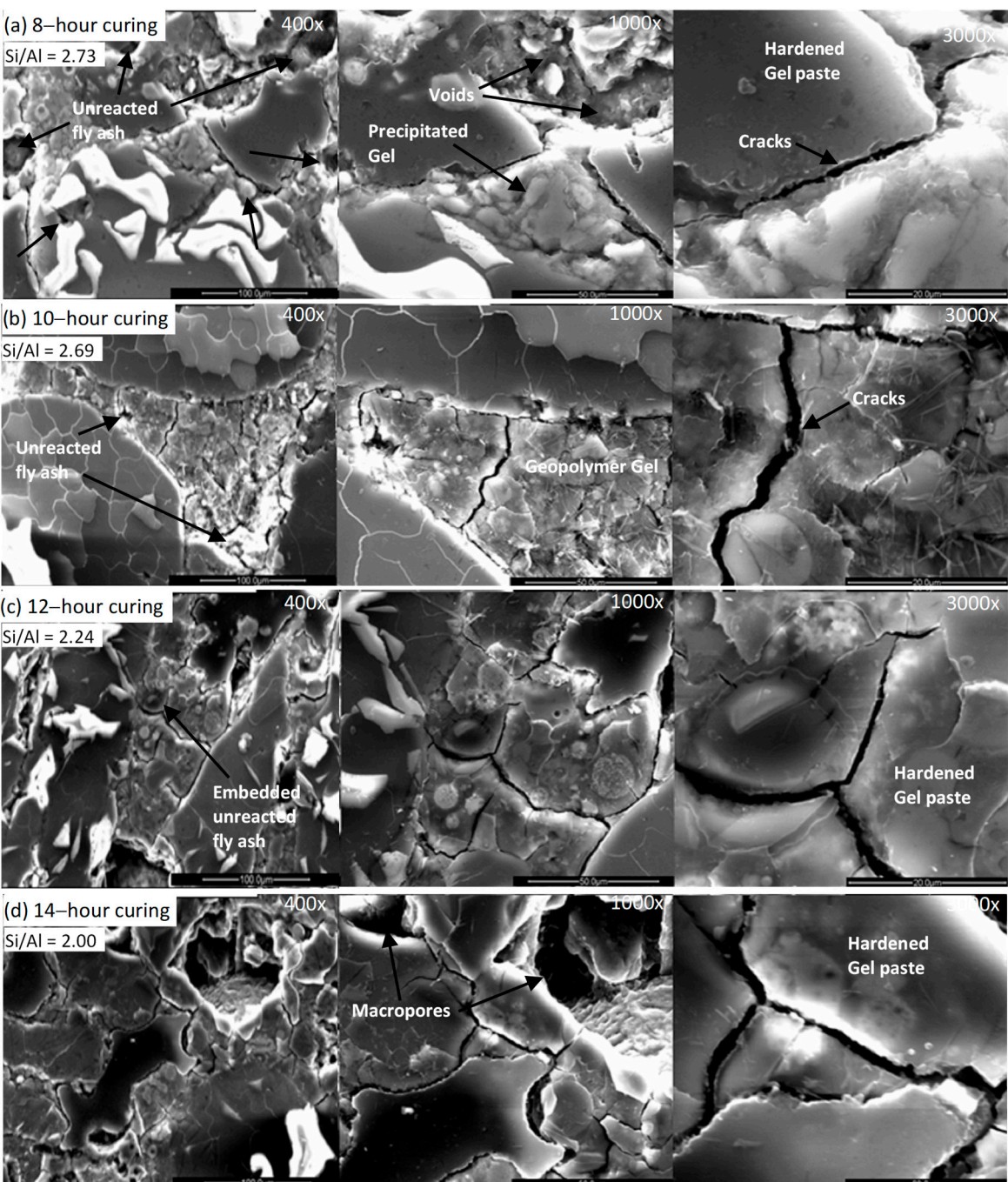

**Figure 7.** The microstructural development of geopolymer mortar (50 mm cube) at 8, 10, 12 and 14 h of heat curing.

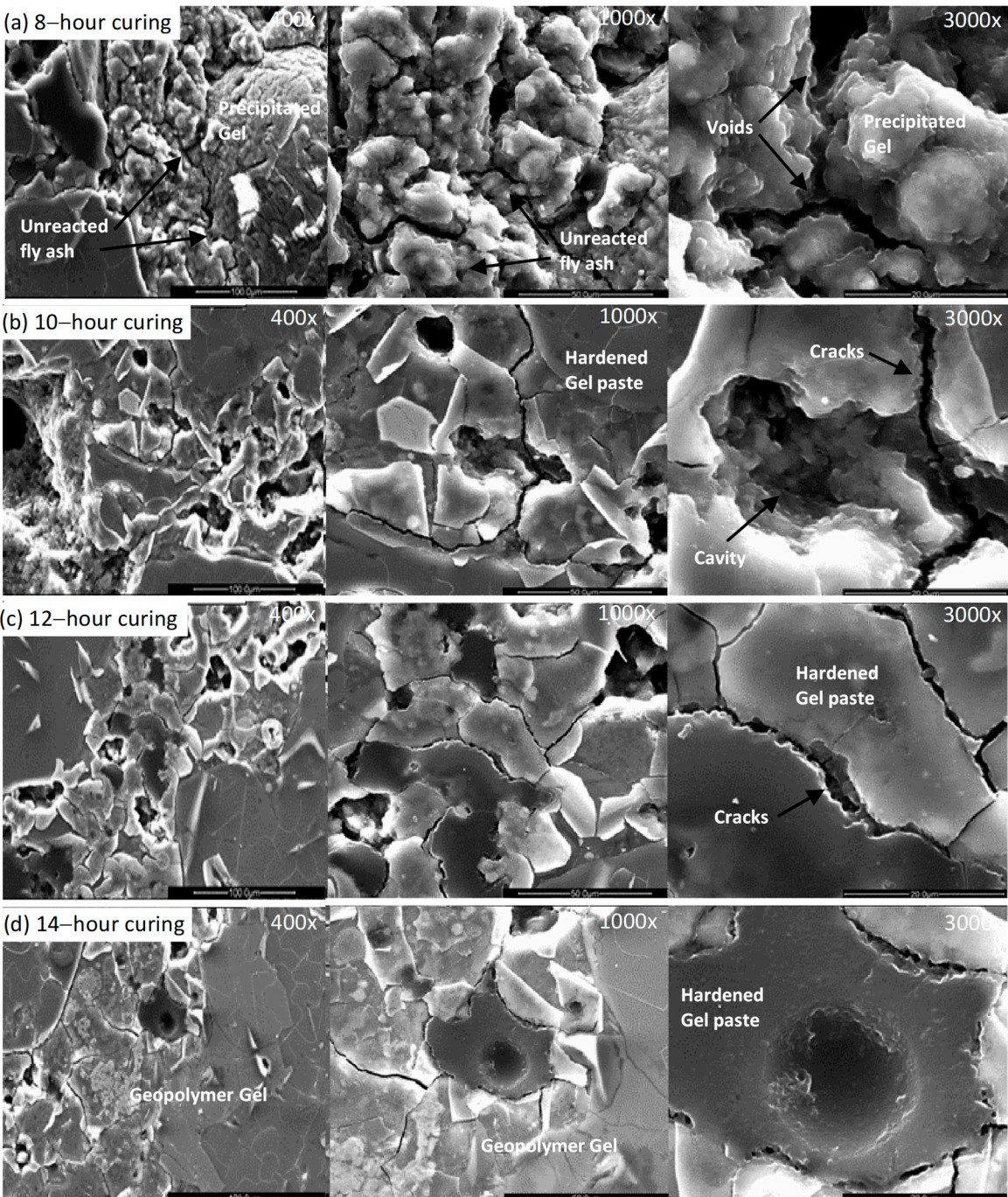

**Figure 8.** The microstructural development of geopolymer concrete (50 mm cube) at 8, 10, 12 and 14 h of heat curing.

## 4.4. Pore-Structure

A comparison of the porosity distribution in the mortar (50 mm cube) and concrete (50 mm cube) between 3 nm to 100 μm is shown in Figure 9. Note that the lower measuring limit of the pore diameter in mercury intrusion porosimeter is 3 nm. The two principal pore structures could be identified based on the pore diameter as mesopores (2–50 nm) and macropores (50–100,000 nm) [43]. When the pore size was below 50 nm, the geopolymer mortar had a higher cumulative pore volume (0.041 mL/g) than the geopolymer concrete (0.024 mL/g). However, in the macropores region, the geopolymer mortar showed a significant total pore volume reduction compared to the geopolymer concrete.

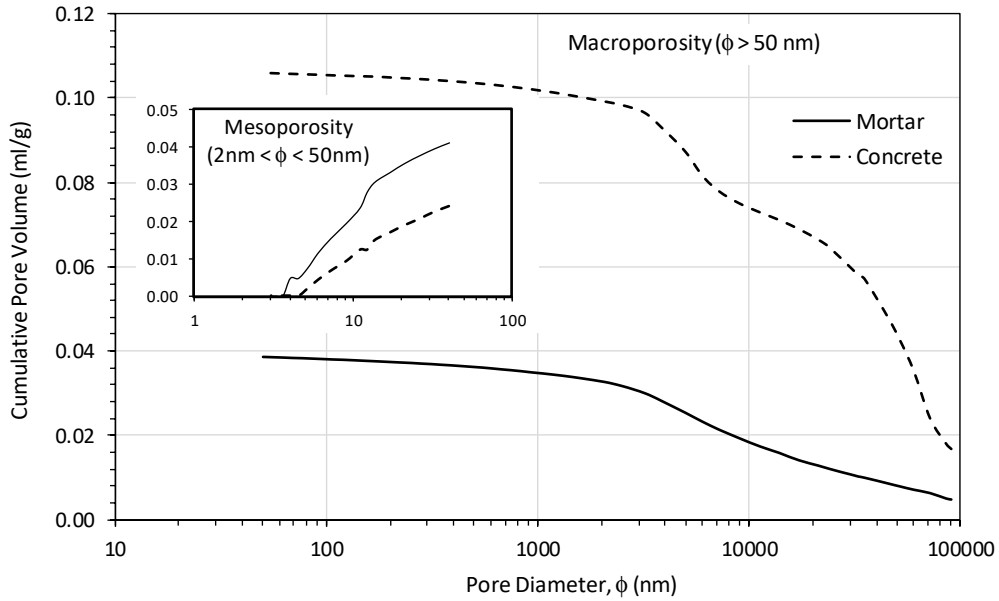

**Figure 9.** The pore size distribution of geopolymer mortar (50 mm cube) versus concrete (50 mm cube).

### 4.5. Pore Interconnectivity

A colour-coded pore distribution together with pore interconnectivity in the 100 mm geopolymer concrete specimen at 4, 8, 12 and 24 h is shown in Figure 10. The pores detected through a CT analysis included partial capillary pores/air voids, while the total voxel count was the main indicator used to acquire the total porosity. Figure 10a–d illustrates a higher heterogeneity in the porosity distribution along the vertical axis. Moreover, the 3D images of geopolymer concrete at each curing age represent the complete 3D pore network, showing the pore connectivity and tortuosity. At 4, 8 and 12 h, the concrete displays a long-interconnected pore network including a range of pores with different sizes. The total voxel after 4, 8 and 12 h of curing, are 6.56%, 5.25% and 4.12%, respectively. However, the porosity of the concrete cured at 24 h (Figure 10a–d) showed significantly lower pore connectivity in the 3D pore network. Indeed, after 24 h of curing, the total voxel count of geopolymer concrete is 1.64%, which is a 76% reduction in porosity compared to the 4 h of curing.

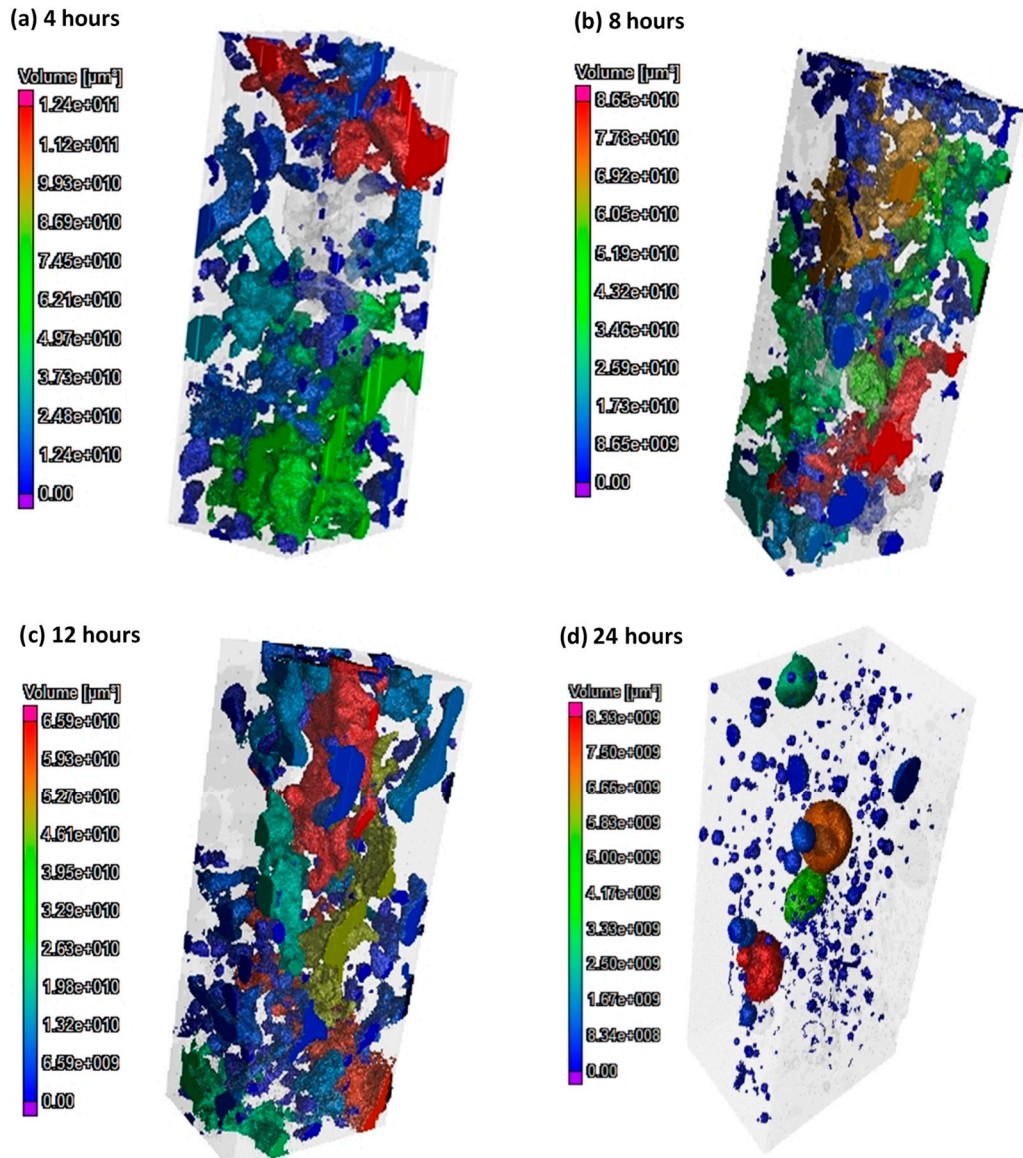

**Figure 10.** Pore interconnectivity and pore volume distribution of 100 mm geopolymer concrete specimen, 4–24 h.

## 5. Discussion

The fly ash geopolymerization is an exothermic polycondensation process which involves alkali activation by a cation in solution. The rapid alkali-fly ash reaction does not allow time for the development of well-structured crystalline phases, and thus hardened fly ash geopolymer concrete is similar to the zeolitic precursors, producing an amorphous, semi-crystalline aluminosilicate gel [44,45].

The microstructure, Figure 7, correlates well with the temperature profile observed for the brown coal fly ash geopolymer mortar, Figure 2. The first stage, 2 to 6 h, can be correlated to the period when the dissolution and speciation equilibrium steps occur as proposed by Duxson et al. (2007) in the conceptual model of geopolymerization. This first stage can also be related to the nucleation and growth stages identified by Duxson et al. (2007). Nucleation is the dissolution of the aluminosilicate material and the formation of polymeric species which are highly dependent on thermodynamic and kinetic parameters. The stage after nucleation is growth. Growth is the condensation-crystallization phase when the nuclei reach a critical size and crystals begin to develop. This process is a structural reorganization which forms the microstructure of the material and the nano-pore distribution which is critical in determining the physical and durability properties of the geopolymer [46,47]. Gelation is

also identified as commencing in this first stage. Dissolution consumes water which is then released during the formation of the gel, with some residual water remaining within pores in the gel. The gel comprises a complex mixture of silicate and aluminate species together with sodium and calcium oxides and is followed by (sodium/calcium) aluminosilicate being released by dissolution into the aqueous phase. This correlates with the temperature profile during this period displaying a slight increase from 105 °C to 108 °C as the nucleation utilizes the energy from the heat curing. The fall in temperature observed, Figure 2, is hypothesised as relating to the energy required during this phase. Moreover, during this stage, the geopolymeric gel matrix begins to form, which corresponds to the strength increase observed from approximately 5 MPa at 4 h to 23 MPa at 6 h.

In the second stage, the temperature profile rose significantly from 108 °C at 6 h to 120 °C at 10 h. The rapid rise of temperature observed in the second stage between 6 to 10 h curing is identified as corresponding to the on-going geopolymerization process, with reduced dissolution, on-going gel formation and growth, and re-organization within the geopolymer matrix. This is reflected in the variation in the Si/Al ratio observed in the specimens. A ratio of 2.73 was noted at 8 h, this decreased to 2.69 at 10 h and further decreased to 2.24 at 12 h. It is hypothesised that this corresponds with gel formation continuing while reorganising occurs in the gel matrix. The $Na^+$ ions act as a structure forming element. The structure of the sodium-aluminosilicate gel contains Si and Al tetrahedrons randomly distributed along the polymeric chains that are cross-linked to provide cavities of sufficient size to accommodate the charge balancing hydrated $Na^+$ ions. It has been noted that geopolymers with higher strengths are more likely to be produced by fly ash having higher $Al_2O_3/Na_2O$ and $SiO_2/Na_2O$ ratios [48,49]. The higher $SiO_2/Na_2O$ weight ratios retard zeolite formation, primarily due to the higher degree of polymerization of soluble silica in the system and which results in the production of more geopolymeric gel [50]. Thus, it is believed that the higher $Al_2O_3/Na_2O$ ratio would add sufficient alumina to react with soluble silica in the system. Thus, a combination of the $Al_2O_3/Na_2O$ and $SiO_2/Na_2O$ ratios influence the compressive strength, and the threshold values for these ratios are required to achieve a specific strength [51].

Furthermore, the dissolution rate starts to reduce during this stage, as water is lost and dissolution ceases. The increased gel formation in this period is reflected in the compressive strength increase to 34 MPa at 10 h. In the final stage, the dissolution process is completed and the evaporation of the remaining water together with a cessation in gel formation and the reorganization step in the geopolymer formation. This corresponds with the slight rise in the temperature profile from 120 °C at 10 h to 123 °C at 14 h. During this stage, the aluminate matrix structure which is attributed as providing the compressive strength is created, with the maximum strength of 42 MPa achieved at 12 h.

The slight rise in temperature at 14 h is indicative of the amalgamation of alumina into the silicate backbone. The results are indicative of a high rate of release and incorporation of aluminium that leads to the production of a gel with a more uniform composition and structure, as observed in Figure 7. This corresponds to the increase in strength observed to the 42 MPa maximum at 12 h. However, at 14 h, the reduction in strength and micro-cracking observed indicated that although there has been incorporation of aluminium into the matrix backbone, the micro cracking resulted in a reduction of strength.

The concrete profiles, Figures 3–5 would suggest a similar mechanism is occurring in the concrete specimens, though the timeframe for the three stages is extended as the specimen size increases. This is supported by a comparison of the temperature profile and the compressive strength. In the 100 mm concrete specimen, the increase in the internal temperature observed in the mortar at 6 h is observed between 12–14 h in the concrete. This is consistent with the compressive strengths which rise from 23.0 MPa to 42.10 MPa in the mortar between 6 and 12 h. In the concrete, where no increase in internal temperature is observed, the strength only increases from 8.75 MPa to 14.25 MPa, but then rises to 30.85 MPa in the period from 12 to 24 h as the internal temperature starts to rise. In the 200 mm and 300 mm cubes, the temperature profiles would suggest that the gelation stage is still underway and that the geopolymerization process is not complete at the center even after 24 h heat curing at 120 °C.

In addition, the condition of the 200 mm and 300 mm specimens observed on demoulding is indicative that geopolymerization is not complete as structural integrity was not achieved.

The increase in strength from 12 to 24 h is also demonstrated by the changes in the pore structure identified by the tomography, Figure 10. During the period from 4 to 12 h, identified as corresponding to stage 1, a reduction from 6.56% to 4.12% in the porosity is observed, while in the period from 12 to 24 h the porosity falls to 1.64%. This is indicative of an increase in gel formation leading to a refinement of the pore structure.

The pore-structure of mortar and concrete specimens (50 mm cubes) also correlate well with the proposed reaction mechanism, Figure 9. Macropores, ranging from 50–200 nm, are formed during the early stage of the geopolymeric reaction. As the gel continues to form, this transforms macropores into mesopores. The presence of macropores larger than 200 nm is characteristic of less reacted geopolymers. This has been reported as primarily occurring in geopolymer pore-structure when synthesized with alkali hydroxide solutions or at low curing temperatures, such as 30 °C, for Class F fly ash geopolymers [43]. The research on these low calcium, Class F fly ash geopolymers has shown that mesopores represent the voids between the geopolymer phases, while micropores primarily are within the gel network. The macropores fill the gaps between unreacted fly ash particles [52]. The geopolymerisation rate controls the quantity of gel formation. In concrete, the gel fills the cracks/voids between unreacted fly ash particles, aggregates the pore space in the matrix, thus refining the size of the pores.

The pores identified for both the brown coal mortar and the concrete are principally in the macropores region, and hence are identified as corresponding to the gaps between unreacted fly ash grains and between the aggregate. This would suggest that little refinement of the pore matrix has occurred within the brown coal fly ash geopolymer materials. The SEM images and the compressive strengths would however, indicate that some degree of refinement has occurred but also show the presence of large macropores in the matrix, correlating with the MIP data, with a significant quantity of macropores being evident. The mesoporosity of fly ash based geopolymer is expected to increase with an increase in the geopolymerization rate, which will result in higher strengths. This correlates with the higher strengths observed for the mortar compared to the concrete in the 4–12 h period and also the temperature profile which indicates that the geopolymerisation progresses more rapidly in the mortar than the concrete.

Overall, the temperature profiles correlate well with the compressive strengths and porosity data and the proposed reaction kinetics for geopolymer formation [44,45]. The results show that the geopolymerisation progresses more rapidly for the mortar specimens than the concrete specimens, with 12 to 14 h providing an optimum curing time for the 50 mm mortar cubes and 24 h being the optimum time for the 100 mm concrete cubes. For 50 mm and 100 mm concrete, the specimens' compressive strengths in excess of 30 MPa can be obtained. However, the larger specimens' structural integrity cannot be achieved. This is attributed to the differential in the temperature profile within the specimens meaning that the geopolymerisation reaction is not complete after 24 h of heat curing at 120 °C in these larger specimens and that longer curing periods are required. However, this may result in micro-cracking in the surface under extended curing periods as the temperature profiles indicate that the reaction in the surface concrete is completed and has reached the point where evaporation of the residual water is likely to occur with longer curing periods. As such, the specimen size and curing time must be taken into consideration in the production of brown coal fly ash geopolymer concrete.

## 6. Further Research Work

The present research study shows that it was feasible to produce a geopolymer concrete with a compressive strength of over 30 MPa from Loy Yang brown coal fly ash. However, the studies raise concerns over the long term performance and commercial viability of large scale specimens. As such, further studies in the area of alkali-activation of brown coal fly ash are needed to assess the optimal curing conditions required for large scale structural elements to achieve the engineering

performance required in order to produce a highly durable geopolymer concrete. In addition, the effect of the optimized elevated curing temperature and the duration of the environmental benefits require assessment.

## 7. Summary and Conclusions

The key findings of this research study can be summarized as follows:

1.  Compressive strength in excess of 40 MPa and 30 MPa can be obtained at 7 days using Loy Yang brown coal geopolymer mortar (50 mm cube) and concrete (100 mm cube) specimens, following curing at 120 °C for 12 h and 24 h, respectively.
2.  The 50 mm brown coal geopolymer mortar and concrete cubes displayed an almost identical temperature profile with an optimal curing period of 12 h. However, the profiles of 100 mm, 200 mm and 300 mm concrete cubes differed significantly, indicating that curing at 120 °C for 12 h is not effective for larger concrete specimens as the internal temperature is not sufficient to promote the geopolymerization.
3.  The temperature profile of brown coal fly ash geopolymer under heat curing can be divided into three stages which correlate well with the proposed reaction mechanism for low calcium class F fly ash geopolymers. However, the time interval of each stage is dependent on the size of the geopolymer specimen.
4.  At the optimum curing time, the dissolution of the fly ash leads to the production of a geopolymer matrix with a more uniform composition and structure. However, the continued heat curing beyond this resulted in micro-cracking in the gel matrix and a reduction in compressive strength.
5.  The specimen size and curing time must be taken into consideration in the mix design stage of brown coal fly ash based geopolymer concrete in order to achieve the optimum compressive strength. Longer curing periods for larger brown coal geopolymer specimens could result in severe micro-cracking on the surface as the temperature profiles indicate that geopolymersiation in the surface concrete is completed and has reached the point where the evaporation of the residual water is likely to occur.

**Author Contributions:** C.G. and R.D. conceived of the presented idea and conducted all experiments. D.W.L. and S.S. verified the experimental and analytical methods, and provided the findings of this work. All authors discussed the results and contributed to the final manuscript.

**Funding:** This project was funded by ARC-ITRH (Australian Research Council-Industrial Transformation Research Hub) research grant (IH150100006) allocated for nanoscience-based construction material manufacturing.

**Acknowledgments:** The authors wish to acknowledge management in Loy Yang power station for the supply of brown coal fly ash. The X-ray facility, Microscopy & Microanalysis facility and scientific & technical assistance provided by RMIT University is further acknowledged.

**Conflicts of Interest:** The authors declare no conflict of interest.

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
