# Peer review of "Effect of Curing Conditions on Microstructure and Pore-Structure of Brown Coal Fly Ash Geopolymers"

_applsci, doi:10.3390/app9153138_

Round 1

Reviewer 1 Report

The paper reports a study on the effect of elevated heat curing on the temperature profile and microstructure development in brown coal fly ash geopolymer mortar and concrete specimens.

I have found that paper to be overall well written, specially Introduction part but important information for better understanding of study aim is missing. General remark is about at least employed temperature (120 ºC). Obtained conclusions could be altered for higher or lower temperatures. 

Some parts of the article can be improved with detailed explanations of materials, better descriptions of procedures, and also adding a few photos from laboratory research.

Line 15: Please, provide a range of temperature values to considerate “elevated temperature curing”.

Lines 33-35: Authors expose that the use of Portland Cement “is responsible for between 5 to 7 % of anthropogenic CO2emissions” without giving a reference and without explaining the importance of CO2emissions. So, please, comment this.

Lines 67-68: “Dirgantara et al. (2017) who reported that the maximum strengths of the geopolymer mortar obtained 68 was 57 MPa”.  Please, specify the type of strength you are talking about and at what age you obtain it.

Line 81: Please, provide again a range of temperatures to consider them “heat curing”.

Lines 82-85: “Sivasakthi et al. (2018) confirmed that micro silica‒fly ash blended geopolymer concrete is thermally stable up to 800 °C and showed that the compressive strength obtained satisfies the requirement of building material with high temperature resistance in modern construction and the ceramic industries”. Could you specify curing age for this assumption?

Line 95: Authors talk about ambient temperature, but this depends on the Country or region. Could you give a range of values for this temperature?

Lines 95-104: Authors give different researches to produce geopolymer concrete for different curing agesand heat curing. Please provide this information also for Palomo et al. (1999).

Lines 126-128: “XRD measurement was performed at 40 kV, Cu Kα = 1.54178 Å wavelength; scanning range 2 theta in 5°–95°. Sample holders were filled using the front-loading procedure”. Why did you decide to use those values and that procedure? Please comment this and give references.

Lines 129-130: “The fly ash surface area was determined using Brunauer Emmett Teller method by N2 absorption” This is same with previous comment.

Lines 133-135: “The alkali activator used contained a mixture of commercially available sodium hydroxide (15 M) solution from Science Supply Australia and sodium silicate solution with a specific gravity of 1.53 and alkaline modulus=2, where Na2O=14.7 %, SiO2=29.4 % and H2O=55.9 % by mass”. Please referee this election.

Line 160: Provide the rpm that uses the vibration table for geopolymer mortar specimens preparation.

Line 164: Same previous comment for geopolymer concrete specimens.

Lines 166-173: Please, provide photos to see the specimens and the monitorization.

Line 204: Change the title with just results because the discussion starts at line 324.

Line 254: There is a mistake in the figure:

Lines 417-424: Authors remark the importance of achieving a curing heat. In this paper, authors have chosen 120 ºC as testing temperature but different temperatures will provide different results. Why did you select this as the curing heat instead of a higher or a lower value? Please comment.

There is a lot of work made, but I have found poor contents on the conclusions.

References must be provided according to the authors guideline.

Author Response

Attached is the addressed comments for Reviewer 1.

Reviewer 2 Report

The manuscript investigated the effect of heat curing on the geopolymeric reaction and strenth evolution in Loy Yang fly ash based geopolymer mortar and concrete. A number of proper analysis technologies were been selected and the results presented systematically. The reviewer thinks the manuscript is qualified for publication in applied sciences, and a few questions are as follows:

1, How did the authors quantify the amorphous and crystalline in Table 2

2, Apart from Al/Si ratio, how do the alkaline elements affect the progress? especially for brown coal ash, The Na content is as high as 6% in Loy Yang ash, and most of them should present in water-soluble species, How does this effect on the geopolymerisation? 

3, The authors conclude that the temperature profile correlated well with the proposed reaction mechanism for class F fly ash geopolymers. what's the proposed reaction mechanism and how does these match?  

Author Response

Attached is the addressed comments for Reviewer 2.

Round 2

Reviewer 1 Report

The same error still appears in Figure 5. Authors should put on the Y-axis:

"Temperature of 300mm concrete specimen (ºC)"

Thanks to the authors for having considered almost all of my requests. In this way the paper becomes clearer and increases its quality.

Author Response

Figure 5 has revised according to the reviewer's comment. Revised Figure is included in the attached word file.
